# The Impact of COVID-19 on the Mental Well-Being of College Students

**DOI:** 10.3390/ijerph19095089

**Published:** 2022-04-22

**Authors:** Natalia Sauer, Agnieszka Sałek, Wojciech Szlasa, Tomasz Ciecieląg, Jakub Obara, Sara Gaweł, Dominik Marciniak, Katarzyna Karłowicz-Bodalska

**Affiliations:** 1Faculty of Pharmacy, Wroclaw Medical University, 50-556 Wrocław, Poland; agnieszka.salek@outlook.com (A.S.); tomasz.ciecielag98@gmail.com (T.C.); jakub.obara@student.umw.edu.pl (J.O.); sara.gavel19@gmail.com (S.G.); 2Faculty of Medicine, Wroclaw Medical University, 50-367 Wrocław, Poland; wojciech.szlasa@outlook.com; 3Department of Drugs Form Technology, Faculty of Pharmacy, Wroclaw Medical University, 50-556 Wrocław, Poland; dominik.marciniak@umw.edu.pl (D.M.); katarzyna.karlowicz-bodalska@umw.edu.pl (K.K.-B.)

**Keywords:** COVID-19, coronavirus, pandemic, college students, mental health, trust in medical authorities, sexuality, financial concern

## Abstract

The COVID-19 pandemic has caused an overall increase in stress and depression in society. The aim of the present research was to evaluate the psychological condition of college students during the COVID-19 pandemic and explore factors influencing their daily functioning. The study focused on four main aspects such as mental well-being, sexuality, concern about financial status, and trust in medical authorities. The study was based on a specially designed survey. The questionnaire was created using Google Forms and shared on social media sites. A total of 630 students participated in the survey, 17 surveys were excluded due to incomplete data and 613 surveys (97.30%) were considered for the final analysis. During isolation, 68.0% of students experienced fear of missing out (FOMO). A total of 73.4% were frustrated due to spending a lot of time in front of a computer. A significant decrease in motivation to study was reported by 78.1% of the respondents. Students showed significantly different attitudes towards sexuality. Concern about the financial situation was reported by 48.7% of respondents. The state of the Polish economy was of concern to 86.4% of respondents. A total of 74.5% of students declared concern about their career development. During the pandemic, 59.0% of respondents became concerned about their health. The attitude towards vaccination was described as positive by 82.5% of the respondents. The percentage of respondents experiencing negative psychological effects relative to the overall epidemiological situation of COVID-19 is troubling. Given the unexpected length and severity of the pandemic, we suggest that students’ concerns be more thoroughly understood and addressed.

## 1. Introduction

The coronavirus disease was characterized as a pandemic by the WHO on 11 March 2020 [1]. Research on mental health during the COVID-19 pandemic is an important area of study that possibly concerns many individuals [2,3]. Time has shown that humanity was not ready for the pandemic’s outbreak. Thus far, the WHO has reported over 486 million cases of COVID-19 and 6.1 million deaths worldwide (April 2022) and, unfortunately, these figures are growing over time [4]. COVID-19 pandemics affected all the population, however, due to the need for rapid progress and self-development young people were one of the most severely suffering groups [5]. Lockdown and social life restrictions led to the changes in three important areas of students’ lives—health, finances and trust in authorities.

Since then, there have been significant changes in many aspects of social life [2,3]. Governments have made decisions about closing borders and public buildings such as universities, thereby forcing students to e-learning [6,7]. All of a sudden, they lost face-to-face contact with other students. Strict restrictions including isolation deteriorated their mental health and increased levels of stress and loneliness [8,9].

The new reality became tough for students, they have been struggling to find their path to follow in it ever since. In Poland alone, over 7.5 million students were deprived of the conventional mode of education [10]. The pandemic had an impact not only on psychological but also on economic aspects [11]. This resulted in changes in the financial status of students who are losing their jobs and access to employment during the pandemic [12,13]. Loss of employment opportunities contributed to financial insecurity which resulted in disturbances in mental well-being [14,15].

Despite vaccination accessibility, we are facing the upcoming sixth wave of COVID-19, reverting to restrictions that may cause anxiety, stress and health concerns [16]. Disturbances in mental well-being need to be thoroughly analyzed to avoid a situation when a significant proportion of current students and soon fully mature people are struggling with mental health issues such as depression or post-traumatic stress disorder. Current research studies were focused more on general fear rather than on multiple factors such as mental well-being, sexuality, concern about financial status and awareness of the current epidemiological situation. The prevalence of young people at high risk of anxiety and low life satisfaction changed significantly across previous waves of the COVID-19 pandemic [17]. Negative changes in lifestyle habits and more time in a mentally passive state while sitting at home may be connected with higher odds of mental ill-health [18].

Concerning the general health of the students, it is important to evaluate their fear about health of themselves and their families and the status of their physical activity (including sexuality). On the other hand, to assess the well-being of the students, we have to question their feelings about missing the contact with the family and friends, their motivation to study, opinion about online studies, social media consumption, computer usage, adequate adjustment of the computer equipment for the online work and worries about the personal development.

The next factor which has to be addressed is the financial status of the students after the onset of the pandemic. To evaluate the problem, one has to question the changes in their own financial state, further the status of their family members and in the end the fear about the financial status of the country in which they are living (here Poland). In this part of the studies there might be some differences between countries due to the government administration of inflation and the general wealth of the country.

The last factor to affect students’ lives is the trust in medical authorities. Here, we include the attitude of the students towards vaccination, compliance to the national restrictions due to COVID-19 and the general state of information about the pandemics.

The study was conducted to assess the changes in all these areas and highlight the most prominent component in each pathological area. The studies may be useful for therapists, administrations and students themselves to avoid further health and social deprivation by targeting and improving the specific areas of live.

## 2. Materials and Methods

### 2.1. Search Method

A rapid review methodology was used to gather evidence [19]. The methodology was appropriate as these reviews aim to provide timely and structured research on rapidly changing situations such as the COVID-19 pandemic [20,21,22]. A systematic computerized search was conducted up to 5 April 2022 in the databases PubMed, Scopus and Web of Science. The articles were collected from sources in which there are a significant number of research articles, particularly on educational institutions. Google Scholar was used to double check search results and search for any relevant studies that were missed during the original database search. The keyword search used terms including COVID-19 and students, COVID-19 pandemic or quarantine, loneliness, mental health and quarantine, financial status and pandemic; these were searched for within the title, abstract, and keyword sections, as a technique used in previous research. Keywords were used regularly and were combined using the advanced search option to ensure that no relevant article was missed. Boolean operators (and, or) were used to increase the number of combined terms and increase the effectiveness of the search scope. The full texts of relevant publications were reviewed for inclusion and exclusion, with the criteria outlined below in Table 1. During the search process, 79 articles were extracted from the databases.

### 2.2. Participants and Data Collection Procedure

Our target group is students from Polish universities who have been in higher education at least since October 2020. For data collection purposes students were asked to complete a self-administered online questionnaire (Google Forms). Participants in the study were generally recruited through social networking services. The data were collected from 30 April 2021 to 5 July 2021. The study was performed on the total group of 630 students—482 females (76.51%) and 148 males (23.49%). Participants had a mean age of 22.29 year. (standard deviation 1.96). Participants were informed that participation was voluntary, and the answers would remain anonymous. The characteristics of the study group are shown in Table 2. Answers were transferred from Google Forms to Microsoft Excel; 17 incomplete or incorrect questionnaires were excluded. Therefore, 613 surveys with full and correct replies were considered for final analysis. The STATISTICA v. 13.3 software by StatSoft Inc., Tulsa, OK, USA was used to determine if selected responses were statistically significant.

Among the students, 185 (30.18%) studied medicine and health sciences, 21 (3.43%) agriculture, veterinary medicine and life sciences, 147 (23.98%) technology and sciences, 260 (42.41%) social sciences and humanities.

### 2.3. Structure of the Survey

A questionnaire consisting of 47 (single and multiple-choice) questions was divided into 5 sections. The first section was about the respondent’s particulars such as age, university, place of residence, and mode of study (1). The rest of the sections helped to collect details about students’ well-being (2), sexuality (3), concerns about the financial status (4) and trust in medical authorities (5). Students could choose between answers on a five-point Likert-type scale starting from “definitely yes” through “yes”, “I don’t know”, “no” ending with “definitely no”.

The section concerning the responders’ information included questions about responders’ gender, age, university, faculty and year of the studies. We asked about the size of the city in which the responder currently stays (<20,000, 20,000–100,000 and >100,000 citizens). Further, we asked about the form of learning in your university (stationary, online or hybrid), about the place of living (family house, lodgings or the dormitory).

The second part of the survey asked about the felling of loneliness, motivation to learn (including competition among students), wanting to return to studying like before the pandemic. We questioned if the students prefer to have the practical lessons remain and lectures online. Well-being was assessed by questioning students’ fear about their own health and their family members’ well-being. We asked about the consumption of tobacco and alcohol. Next, we asked about the contact with peers and family. The questionnaire asked about the time spent on social media and generally online. Next, we moved to questioning the responders about the fear of missing out (FOMO) and the need for help from the specialists.

Questions about sexuality included the changes in the libido, frequency and satisfaction of sexual activity. One question concerned the status of participants’ relationships. Next, we asked if there had been any changes in the relationship with the partner. Aside from the sexual activity and relationship, we asked about the changes in physical activity, hobby and free time.

The fourth part of the survey asked about the financial aspect of the pandemic including the lack of money for the online learning hardware. We also asked about the concerns about the financial status of Poland, future career development and the changes in the financial stability of the responders and responders’ families. Moreover, we questioned the changes in the employment status and monthly income.

Finally, the last part was associated with the trust in medical authorities. To do so, we asked about the source and frequency of seeking information related to COVID-19. Further, we moved to the attitude of the Polish students to the vaccination. We questioned the type of vaccine that was the most popular among students. The last question was connected with adverse events following immunization.

The most interesting results of the survey are presented in the figures.

### 2.4. Chi-Squared Test

The chi-squared χ^2^ test was used to determine if variables depend on each other, compare the expected and observed values of these variables and the standard deviations between them. The examination of the differences allowed determination of if they are major differences and conclude that one variable has an impact on another one [23]. The obtained statistical tests and degrees of freedom allowed tracing of statistical significance. A contingency table was developed with the distribution of variables, which would make it possible to determine how many students answered “definitely yes” to a pair of questions in comparison with each other and to the rest of the different combinations of answers. The last step was to select the responses in which the differences between the groups were statistically significant (*p*-value not greater than 0.05).

### 2.5. Meta-Analysis and Odds Ratio (OR)

Meta-analysis is the statistical combination of results from two or more separate studies. For the needs of meta-analysis, answers have been converted into dichotomic responses. “Definitely yes” and “yes” were counted as “positive” and the rest of the answers were counted as “negative”. In the presented research, these analyses were used to check how different variables influence a particular variable. Those with a *p*-value lower than 0.05 were selected. It was also necessary to create a contingency table. Odds are defined as the ratio of success to failure. In this study, success was assumed as a “positive” and failure as a “negative” response. The odds ratio (OR) is the ratio of odds in the tested variability of two groups, i.e., the tested (exposed) and reference (non-exposed) group [24].
OR = P(A)1 − P(A)P(B)1 − P(B)(1)
where:

P(A) = “positive” in the tested group;

1 − P(A) = “negative” in the tested group;

P(B) = “positive” in the reference group;

1 − P(B) = “negative” in the reference group.

A higher OR value means a greater chance that more variables have an impact on the analyzed aspect. OR > 1 signifies that the odds are greater in the tested group than in the reference group. The aim was to determine how many variables influence the examined aspect. This is illustrated in forest plots containing OR, *p*-value, confidence interval and its cutoff points.

## 3. Results

### 3.1. Well-Being of Students

Lockdown introduced in connection with COVID-19 resulted in 81.1% of our respondents lacking contact with their peers and induced 79.3% of them to spend more time on social media. During isolation, 68.0% of students suffered from fear of missing out (FOMO)—especially those who declared spending more time on social networking sites (*p* < 0.0001). This group of students also lacked contact with their peers (*p* < 0.0001) and wanted to come back to non-remote learning (*p* < 0.0001). Furthermore, spending more time on social media was associated with the frustration of spending more time in front of a computer (*p* < 0.0001). In general, 62.5% of respondents wanted to come back to non-remote classes and 73.4% of them were frustrated due to them spending too much time at the computer. This frustration was characteristic for remote learners (*p* = 0.0166) who lacked contact with people of similar age (*p* < 0.0001) and resulted in reduced motivation to study (*p* < 0.0001), a subjective feeling of mental deterioration requiring the intervention of a specialist (*p* = 0.0003) and resulted in more frequent checking of information about the pandemic in Poland (*p* = 0.0009). A significant reduction in motivation to study was reported by 78.1% of respondents. It was also correlated with FOMO (*p* < 0.0001), which in turn increased the need for contact with equals (*p* < 0.0001) and the impression of competition among students (*p* = 0.0006). In the study group, 19.4% of students felt an increase in competition among their colleagues and 36.4% of them indicated that their poor mental health required specialist intervention. Students who participated in the study were also asked about how they spend free time during the pandemic. A total of 34.1% of them spend it on physical activity and 45.9% spend it on hobbies. Respondents who declared engaging in a hobby (46.0%) seem to be also physically active (*p* = 0.0300) and less interested in coming back to non-remote studies (*p* = 0.0275). The presented study indicates that physical activity seems to have a positive impact on the mental health of students. Physically active students tended to smoke less (*p* = 0.0497), and they felt less lonely (*p* = 0.0083). The impact of a feeling of loneliness on other aspects of functioning and well-being of students was examined over the course of a meta-analysis (Figure 1).

### 3.2. Sexuality

Sexual intercourse was more interesting for 21.5% of students questioned in the survey, 16.6% of them were more sexually active during the pandemic and 14.2% stated that they were more satisfied with their sex life. On the other hand, 17.6% of respondents were less interested in sexual activity, 19.6% of them declared reduced sexual activity and 16.3% of them reported lower sexual satisfaction. Only 6.5% of respondents’ relationships ended during the pandemic. A total of 12.3% of students declared entering into a new relationship during the lockdown. A total of 14.2% of respondents declared that their relationship with their current partner worsened, whereas 20.0% of them improved their contact with a partner. The chi-squared test confirms that the sexual activity of students may be affected by worries about their health (*p* = 0.0004). The analysis of responses shows that in the group of respondents who were definitely worried about their health, interest in sexual activity was definitely reduced, but in the group that describe themselves as rather worried, interest in intercourse has increased. In the group declaring lack of health worries, interest in sexual intercourse has not changed. A meta-analysis of the data showed that people more interested in sexual activity also tended to smoke more (*p* = 0.0200), suffered from FOMO (*p* = 0.0325) and felt frustrated with their computers not being adapted to the requirements of educational platforms (*p* = 0.0141).

### 3.3. Concerns about Financial Status

The personal financial situation was disturbing for 48.7% of respondents. During the pandemic, 76.0% of them lost their jobs and 26.4% declared they had less money to spend monthly compared to the time before the lockdown. Students were also worried about the financial status of their families (37.2%). The chi-squared test and the analysis of responses show that people who were not worried about their finances (25.4% of respondents) were also not worried about their own health (*p* = 0.0002). The meta-analysis of the data proves that apprehension about personal financial status increases the number of tobacco products smoked (*p* = 0.0214) and the amount of alcohol consumed (*p* = 0.0500). However, the increase in alcohol consumption in the general population was not statistically significant (*p* > 0.05) after the pandemics started. This group of students observed an increase in competition among students (*p* = 0.086), the need for more contact with peers (*p* = 0.0054) and worries about the economic situation of their families (*p* < 0.0001) and of the entire country (*p* < 0.0001). Furthermore, many of them declared deterioration of physical health requiring specialist help (*p* < 0.0001). The condition of the Polish economy was bothering 86.4% of respondents. According to the meta-analysis, this concern was distinctive for students with insufficient contact with their peers (*p* < 0.0001), spending more time on social media (*p* = 0.0113) and with decreased motivation to study (*p* < 0.0001). Respondents who worried about the Polish economy checked information about COVID-19 more often (*p* = 0.0007) and they expressed the concern that their mental health deteriorated enough that they had to seek the help of a specialist (*p* = 0.0003). Many students were worried about their future careers, which negatively influenced their psychological condition (Figure 2). In general, 74.5% of students declared career development concerns.

### 3.4. Trust in Medical Authorities

During the pandemic, 59.0% of our respondents began to worry about their own health. This misgiving affected many areas of their life-among other things, prompted them to check information regarding the pandemic situation in the country (Figure 3). A total of 41.2% of respondents declared checking information about COVID-19 every once in a while; 18.7% of students checked information very rarely or never, 20.8% did it daily and only 2.7% of students checked it a few times a day. Announcements published by sanitary and epidemiological services and by the World Health Organization seem to be the most popular sources of information, another popular option were posts by medical professionals. In the study group, 39.3% of students were vaccinated against COVID-19. When asked about attitudes towards vaccinations, 82.5% of respondents described them as positive, for 10.4% it was hard to tell and 7.0% of the students referred to them as negative. Certain dependencies were observed for the answer to this question among students on various types of faculties (*p* = 0.0002). Students of medical and health-related faculties were likely to describe their attitude towards vaccination against COVID-19 as very positive. Additionally, this group revealed a strong negative correlation with the group of respondents who were not sure about their opinion. Students of agricultural, veterinary and natural sciences seem to have a rather positive attitude toward vaccination. Opinions of science students and students of social and humanities faculties about vaccines against COVID-19 were rather negative. Furthermore, those two groups of students showed a positive correlation with the group of respondents who were not sure about their opinion. The type of faculty had no impact on the assessment of the quality of healthcare by students, but it was influenced by family history of the disease (*p* = 0.045). Generally, 77.1% of respondents declared that their families were receiving healthcare during the pandemic. The group of students in whose close family someone had fallen ill (48.4%) tended to assess health care professionals rather positively or definitely negatively. In the group where a member of the extended family was ill (13.7%), the assessments were rather negative or it was hard to tell for them. In the group without a family history of the disease (37.8%), assessments tended to be definitely positive or it was hard to tell for them.

## 4. Discussion

The COVID-19 pandemic has introduced many changes into the life of society, such as the necessity to adhere to the principles of social distancing, remote learning, and the loss of jobs affecting many people. Isolation has significantly affected the mental comfort of young people and forced them into solitude. The main aim of the study was to assess the impact of the COVID-19 pandemic on the mental health of students and explore factors influencing their daily behavior.

In a review of studies conducted in Poland before the pandemic, the prevalence of FOMO was estimated to be 67% (mild level of FOMO) [25]. Furthermore, another review on fear of missing out focused on the student population showed a strong correlation with the increased use of social media [26]. The present work shows that FOMO has not disappeared during the pandemic, even in a situation of social distancing at home. A study conducted by Hayran et al. [26] revealed that people experiencing FOMO tended to stay up later at night due to a desire to catch up on social media, which resulted in lack of sleep, difficulty concentrating and lower motivation to complete daily tasks. Interestingly, high educational background is a risk factor for COVID-19 informational FOMO, which places students in a group at higher risk of its effect [27]. Baker et al. [28] found that experiencing higher levels of FOMO was associated with more depressive symptoms, less mindful attention, and more physical symptoms. It was proved that, during COVID-19 lockdown, FOMO levels have strengthened attitudes toward online communication [29]. The pandemic forced students to study remotely and indirectly influenced their increased presence in social media. Therefore, it is not surprising that social media platforms have seen radical leaps in popularity. In the early stages of the pandemic, it has been reported that the total internet hits have surged by between 50% and 70% [30]. The use of social networking apps may have a detrimental effect on young adults’ mental health by inducing greater sense of loneliness and consequently fear of missing out (FOMO) [31]. Our study indeed shows a significant correlation between students spending more time on social media and FOMO. Furthermore, the number of young people who started to feel anxiety, loneliness, and stress during isolation are increasing [32,33,34,35]. Over 65% of respondents declared that they feel lonely, and more time spent in social media aggravated loneliness. Restrictions imposed on students because of isolation also harmed their motivation to study [36,37]. Recent studies showed that students of faculties unrelated to IT must increase the level of their skills to benefit from e-learning, the lack of adequate computer skills in this field and deficiency of proper training were additional obstacles to successful learning [38,39]. In our study, most of the students had no experience in the IT sector; we found that, with increasing frustration about computer problems, their study motivation decreased. Lockdown has limited opportunities for many outdoor activities and many people have been forced to abandon their hobbies. A study from Jordan found a significant correlation between the frequent practice of hobbies and mental well-being; people who engaged in their hobbies felt less lonely [40]. Similar tendencies have been noted in the British population, confirming that various interventions, such as learning a new hobby, can provide a distraction and help combat loneliness [41]. Several studies show that, during the constraints of the COVID 19 closure, physical activity was positively associated with psychological well-being [42,43,44,45]. These data are consistent with our findings; physically active students were less likely to smoke cigarettes and feel less lonely. During the pandemic, the majority of university students from different countries had reduced levels of physical activity: walking, moderate, vigorous and total [46,47]. A study performed on Ukrainian students revealed that the physically inactive group had higher anxiety and depression levels than the physically active group [44]. It clearly shows that limitations during isolation forced students to change their lifestyles and negatively affected their well-being.

In the era of lockdown orders and social distancing guidelines, students experienced drastic changes in their way of life, with significant implications for leisure activities, including sex. Research confirms the importance of the role of sexual pleasure as beneficial for health and well-being [48,49,50,51]. Stringent COVID-19 restrictions forced students to stay at home and limited their recreational sexual behaviors; for many of them it was a period of celibacy. Stress and constant health concerns had a significant impact on libido. Our results suggest that students who were paralyzed by stress associated with health concerns had no interest in sex, but what is interesting is that those with moderate levels of stress (possibly chronic stress related to COVID-19) reported a higher level of interest in sex. This result is also consistent with a prior study suggesting that stress and loneliness were linked to trying new things [52]. In the face of drastic changes in their daily lives, many students have adapted their sex lives in creative ways; constant stress could potentially prompt more sexual adaptation to fulfill psychological needs or relieve negative mood states [52]. During the lockdown, there was an increase in downloads of dating apps and online pornography searches [53,54]. Integration of the internet and digital platforms in the sex life of students, suggests that when opportunities for partner sex are limited; young people more often decide to take solo and online activities as an adaptation to the new situation.

During the pandemic, the economic situation in Poland deteriorated significantly, which affected students’ finances. In the study sample, 49% of respondents were worried about their finances and 76% of them lost their jobs. Dramatic changes in the financial status had a significant impact on the mental well-being of students. A similar study conducted at a British university suggests that economic concerns might adversely affect mental and physical health in student populations [55]. Concerns about financial status were often associated with emotional problems, poorer social functioning and more negative perceptions regarding their well-being [56,57,58,59]. Furthermore, the lockdown caused by the COVID-19 pandemic contributed to job losses among many students; higher job insecurity was also related to worse mental health [13,60,61,62]. A study conducted by Goudeau et al. [63] proved that the period of school closures may accelerate the reproduction of social inequalities in educational achievement. The findings of the presented study show that the majority of participants who expressed concern about their financial status had a higher level of distress about their future career development and were more anxious about the economic situation in the country. Financial stress is a crucial factor that affects student anxiety; lack of financial resources can lead to increased stress through poor nutrition and housing [64]. The presented study found that students who lost their jobs were more worried about finances and declared health concerns; additionally, motivation to study in this group was reduced. The relationships found between financial stress and mental well-being indicate that financial stress is relevant for college students and may lead to negative outcomes.

The present study has a limitation—sample size for our survey (630) was not greater than the ones used in some other studies. However, smaller sample studies concerning similar problems support our results; therefore, we assume our analyzed group may represent the general population relatively accurately. There have already been published studies on a similar size of population with reliable results [28,65,66]. First, our result of the decreased well-being of students is supported by Elmer’s study, which revealed higher levels of stress, anxiety, loneliness, and depressive symptoms during the COVID-19 pandemic [65]. A study performed on about 250 Swiss individuals supported our result, claiming that the stress, anxiety, loneliness, and depressive symptoms worsened during the COVID-19 pandemics [65]. Additionally, Alradhawi et al. [67] in their review focus on students as the potentially susceptible group to disrupt their well-being by the disruption of daily routines. The second factor, which worsened during the COVID-19 pandemics was the sexuality of the students. A study performed on 212 USA students revealed that the sexual activity of American students decreased dramatically for over 57% of reports [68]. Interestingly, a study by Holland et al. [69] warns that the level of sexual harassment in higher education has to be monitored even during pandemic times. Additionally, the narrative review by Eleuteri et al. [70] focuses on the reduced sexual activity as the trigger to pornography consumption among young people. The third problem of students are the concerns about financial status, which according to our studies increased among polish students. Similarly, study performed on New York city students (10,000 sample size) obtained the same result as our study, showing that the students experienced the anxiety/depression and the financial instability due to the pandemics [71]. The last factor that was analyzed in this study was the trust in medical authorities. Here, the highest differences between different countries-based studies are expected. Interestingly, the tendencies were also similar. For instance, most of the 1581 responders from the Romanian students survey had a positive attitude towards vaccination and a similar tendency was observed among polish students [72]. In Saudi Arabia, most of the 1232 participants complied with precautionary practices in force during the COVID-19 pandemic, which was generally observed among polish students as well [73].

Digital media has become an increasingly important information source for health and crisis communication during the outbreak of the pandemic [74,75,76]. Survey participants most often looked for information from reliable sources, such as WHO and medical professionals on social media, which may be related to the fact that most of them study at medical and health faculties. Furthermore, in this group of students, attitudes toward vaccination against COVID-19 were more positive than in participants from other faculties. The level of medical knowledge may be associated with the willingness to adhere to the restrictions.

## 5. Conclusions

Our results highlight that the mental health of university students in Poland is significantly affected in the face of a public health crisis. Students require help and support from the university, society and family. The positive outcome of the study was that physical activity may reduce loneliness, increase mental well-being and reduce smoking tendencies. This study may be a suggestion for the government and universities to improve their collaboration with students in order to address mental health issues and related stress, loneliness, and decreased motivation to study, and provide students with better quality and availability of psychological services, educational platforms, increase motivation for physical activity, and improve medical knowledge of students majoring in subjects not related to public health. Our findings help to specify the areas which disrupt the whole of students’ lives and therefore may be used in the social campaigns fighting the mental, social and financial side effects of COVID-19.

## Figures and Tables

**Figure 1 ijerph-19-05089-f001:**
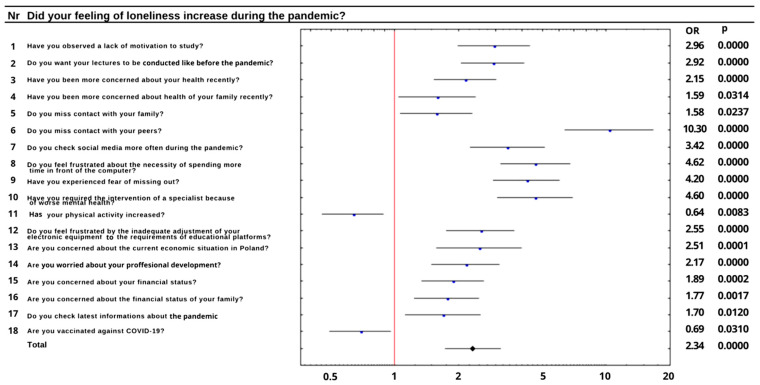
The impact of the feeling of loneliness on other variables.

**Figure 2 ijerph-19-05089-f002:**
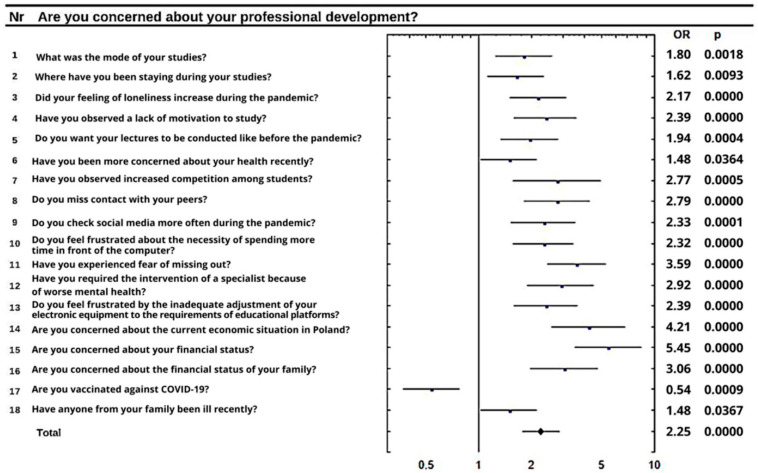
The impact of anxiety about personal development on other variables.

**Figure 3 ijerph-19-05089-f003:**
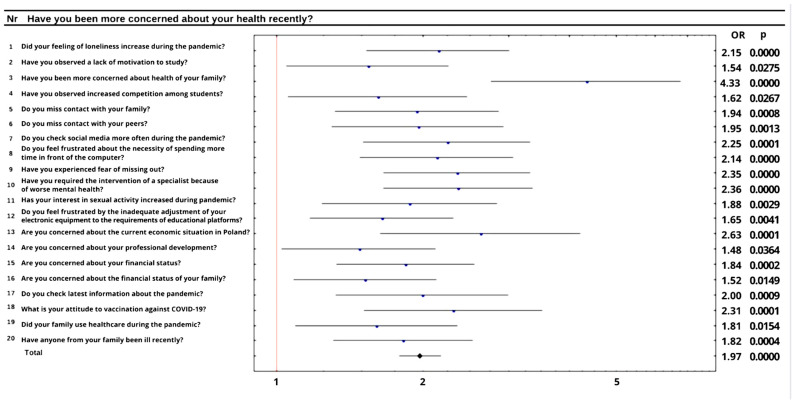
The impact of health anxiety on other variables.

**Table 1 ijerph-19-05089-t001:** Inclusion and exclusion criteria.

Inclusion Criteria	Exclusion Criteria
(a) Articles indexed in Google Scholar	(a) Pre-printed articles
(b) Articles from reliable journals	(b) Studies in other languages with no English translation.
(c) The journal/article was related to COVID-19 (d) All countries	(c) Studies dealing with other viruses.

**Table 2 ijerph-19-05089-t002:** Characteristics of students group.

Sex	Male	Female
Number participants	148	482
Vaccination (%)	44	38
Age (y.o. +/− SD)	22.69 ± 1.64	22.18 ± 2.03
Year of the study (first half/second half)	77/71	341/141

## Data Availability

Not applicable.

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
