# Peer review of "The Impact of COVID-19 on the Mental Well-Being of College Students"

_ijerph, 2022, doi:10.3390/ijerph19095089_

Round 1

Reviewer 1 Report

An interesting and timely paper. I am curious about whether or not participant gender information was factored into analyses. I would also be interested to see a bit more discussion of and references related to FOMO.

There are places throughout where some minor copy editing for English style will improve the paper.

Author Response

Thank You for Your interesting observations. Here the original questions and answers to Your suggestions.

An interesting and timely paper. I am curious about whether or not participant gender information was factored into analyses.

 We are very grateful for the kind review. As you suggested, we have included this information in a new table “Characteristics of students group” which is a table 2 now. We hope that we included all the required information.

 I would also be interested to see a bit more discussion of and references related to FOMO.

 Thank you for your comment, we have added more information related to FOMO in discussion:

 “A study conducted by Hayran et al. revealed that people experiencing FOMO tended to stay up later at night due to a desire to catch up on social media, which resulted in lack of sleep, difficulty in concentrating and lower motivation to complete daily tasks. Interestingly, high educational background is a risk factor for COVID-19 informational FOMO, which places students in a group at higher risk of its effect [27]. Baker et al. found that experiencing higher levels of FOMO was associated with more depressive symptoms, less mindful attention, and more physical symptoms [28]. It was proved that during COVID-19 lockdown, FoMO levels have strengthened attitudes toward online communication [29]. The pandemic forced students to study remotely and indirectly influenced their increased presence in social media. Therefore, it is not surprising that social media platforms have seen radical leaps in popularity. In the early stages of the pandemic, it has been reported that the total internet hits have surged by between 50% and 70% [30]. The use of social networking apps may have a detrimental effect on young adults' mental health by inducing greater sense of loneliness and consequently fear of missing out (FOMO) [31].”

 There are places throughout where some minor copy editing for English style will improve the paper.

 Thank you for the suggestion, we have checked the manuscript a few times and now it should not contain any spelling nor grammar mistakes.

Reviewer 2 Report

The search strategy and its results are not implemented (or described). The only publication really on the topic that is mentioned in the theoretical background is this one:

Yatsuya, H.; Ishitake, T. Health of university students under job and financial insecurity during COVID-19 368 pandemic. J. Occup. Health 2021, 63, e12223, doi:10.1002/1348-9585.1222

It is not clear what theoretical underpinnings operationalized the hypotheses and hypotheses into the questionnaire, and why the core questionnaire domains were chosen in the first place. The relevant literature is only described in the discussion (17ff.), implying that targeted searches were done. 

The questionnaire itself was unclear in its operationalization (why these particular domains (finance, sex, mental well-being, trust in medical authorities???) were investigated), resulting in a national study with no possibility of international thematic comparison using the same collection instrument. This deficiency was also reflected in the choice of keywords, which do not capture the essence of the article. Why don't the authors mention other selected areas in the keywords, why only mental health (or do they include sex, finances trust in doctors)?

A more detailed description of the research population (which universities, which faculties) is missing - or were the questions formulated in such a way that no further socio-demographic data were necessary?

Perfectly statistically processed results without a deeper description of the problem identification and operationalization of the hypotheses into the actual data collection instrument cannot, I stress again, cannot replace or circumvent the validity of the data collection.

As an author, if I decide to take on a relatively sterile topic (like the topic of the impact of C-19 on mental health), I must at least have a perfect methodology, especially for a survey.

The Discussion is well done, maybe if the authors would break away from it and fix the whole introduction and methodology for a new paper, it could be accepted (though in my opinion in a less prestigious journal).

In the Conclusion, the authors repeat the results from the Results and Discussion. For impact articles, the rule of thumb is - don't repeat anything! The last paragraph of the Conclusion is useful: students require help and support from the university, society and families and further: it would be useful to elaborate this message, not only in general terms, but really intertwined with the results.

Author Response

Thank You for Your interesting observations. Here the original questions and answers to Your suggestions.

The search strategy and its results are not implemented (or described). The only publication really on the topic that is mentioned in the theoretical background is this one:

Yatsuya, H.; Ishitake, T. Health of university students under job and financial insecurity during COVID-19 368 pandemic. J. Occup. Health 2021, 63, e12223, doi:10.1002/1348-9585.1222

It is not clear what theoretical underpinnings operationalized the hypotheses and hypotheses into the questionnaire, and why the core questionnaire domains were chosen in the first place. The relevant literature is only described in the discussion (17ff.), implying that targeted searches were done.

The questionnaire itself was unclear in its operationalization (why these particular domains (finance, sex, mental well-being, trust in medical authorities???) were investigated), resulting in a national study with no possibility of international thematic comparison using the same collection instrument. This deficiency was also reflected in the choice of keywords, which do not capture the essence of the article. Why don't the authors mention other selected areas in the keywords, why only mental health (or do they include sex, finances trust in doctors)?

 Thank you for the comment, we have added the required search strategy and described the theoretical background of the study in more detail as presented below:

“2.1. Search method

A rapid review methodology was used to gather evidence [19]. The rapid review methodology was appropriate as these reviews aim to provide timely and structured research on rapidly changing situations such as the COVID-19 pandemic [20–22].A systematic computerized search was conducted up to 5th April 2022 in the databases PubMed, Scopus and Web of Science. The articles were collected from sources in which there are a significant number of research articles, particularly on educational institutions. Google Scholar has been used to double check search results and search for any relevant studies that were missed during the original database search. The keyword search used terms including COVID-19 and students, COVID-19 pandemic or quarantine, loneliness, mental health and quarantine, financial status and pandemic; these were searched for within the title, abstract, and keyword sections, as a technique used in previous research. Keywords were used regularly and were combined using the advanced search option to ensure that no relevant article was missed. Boolean operators (and, or) were used to in-crease the number of combined terms and increase the effectiveness of the search scope. The full texts of relevant publications were reviewed for inclusion and exclusion, with the criteria outlined below in Table 1. During the search process, 79 articles were extracted from the databases.

Table 1. Inclusion and exclusion criteria

Inclusion criteria

Exclusion criteria

a) Articles indexed in google scholar

a) Pre-printed articles

b) Articles from reliable journals

b) Studies in other languages with no English translation.

c) The journal/article was related to COVID-19

d) All countries

c) Studies dealing with other viruses.

Moreover, we have doubled the references list to improve the study. Now we hope that both our search criteria are acceptable and the theoretical background is comprehensive.

Moreover, we have built the whole new section about the construction of the questionnaire:

“A questionnaire consisting of 47 (single and multiple-choice) questions was divided into 5 sections. The first section was about the respondent’s particulars such as age, university, place of residence, and mode of study (1). The rest of the sections helped to collect details about students’ well-being (2), sexuality (3), concerns about the financial status (4) and trust in medical authorities (5). Students could choose between answers on a five-point Likert-type scale starting from “definitely yes” through “yes”, “I don’t know”, “no” ending with “definitely no”.

The section concerning the responders’ information included questions about re-sponders’ gender, age, university, faculty and year of the studies. We asked about the size of the city in which the responder currently stays (<20000, 20000-100000 and >100000 citizens). Further we asked about the form of learning in your university (stationary, online or hybrid), about the place of living (family house, lodgings or the dormitory).

The second part of the survey asked about the felling of loneliness, motivation to learn (including competing with the others), feeling like to get back to studying like before pandemics. We questioned if the students prefer to have the practical lessons stationary and lectures online. Well-being of the students were assessed by questioning students’ fear about their own health and their family members well-being. We asked about the consumption of tobacco and alcohol. Next, we asked about the contact with the peers and family. The questionnaire asked about the time spent on social media and generally online. Next, we moved to questioning the responders about the fear of missing out and the need for help from the specialists.

Questions about sexuality included the changes in the libido, frequency and satis-faction of the sexual activity. One question concerned the status of the relationship. Next, we asked if there were any changes in the relationship with the partner. Aside from the sexual activity and relationship we asked about the changes in physical activity, hobby and free time.

Fourth part of the survey asked about the financial aspect of the pandemics including the lack of money for the online learning hardware. We also asked about the concerns about the financial status of Poland, future career development and the changes in the financial stability of the responders and responders’ families. Besides, we questioned the changes in the employment status and monthly income.

Finally, the last part was associated with the trust in medical authorities. To do so, we asked about the source and frequency of seeking information related to COVID-19. Further we moved to the attitude of the Polish students to the vaccination. We questioned the type of vaccine that was the most popular among students. Last question asked about the adverse event following immunization.

The most interesting results of the survey were presented in the figures.”

According to the rationalization about the choice of questions and areas of interest to the questionnaire, we choose the most important areas of the students’ lives, which is now described in detail in the introduction:

“Concerning the general health of the students it is important to evaluate their fear about health of themselves and their families and the status of their physical activity (including sexuality). On the other hand, to assess the well-being of the students we have to question their feelings about missing the contact with the family and friends, their motivation to study, opinion about online studies, social media consumption, computer usage, adequate adjustment of the computer equipment for the online work and worries about the personal development.

The next factor which has to be addressed is the financial status of the students after the onset of the pandemics. To evaluate the problem, one has to question the changes in their own financial state, further the status of their family members and in the end the fear about the financial status of the country in which they are living (here Poland). In this part of the studies there might be some differences between countries due to the government administration of inflation and the general wealth of the country.

The last factor to affect students’ lives is the trust in medical authorities. Here, we include the attitude of the students towards vaccination, compliance to the national re-strictions due to COVID-19 and the general state of information about the pandemics.

The study was conducted to assess the changes in all these areas and highlight the most prominent component in each pathological area. The studies may be useful for therapists, administration and students themselves to avoid further health and social deprivation by targeting and improving the specific areas of live.”

Additionaly, we added keywors related to topic of our study “trust in medical authorities, sexuality, financial concern”.

We truly hope, that now the background about our survey study is clear and the whole manuscript was improved.

 A more detailed description of the research population (which universities, which faculties) is missing - or were the questions formulated in such a way that no further socio-demographic data were necessary?

According to the Reviewer’s comment, we have included this information in a new table “Characteristics of students group” which is a table 2 now. We hope that we included all the required information. Moreover, aside from the description of the questionnaire in the manuscript, we also provided the original document to the Editor.

Perfectly statistically processed results without a deeper description of the problem identification and operationalization of the hypotheses into the actual data collection instrument cannot, I stress again, cannot replace or circumvent the validity of the data collection.

As an author, if I decide to take on a relatively sterile topic (like the topic of the impact of C-19 on mental health), I must at least have a perfect methodology, especially for a survey.

The Discussion is well done, maybe if the authors would break away from it and fix the whole introduction and methodology for a new paper, it could be accepted (though in my opinion in a less prestigious journal).

 We are grateful for the comment, as mentioned earlier, we have described the data collection process (i.e. questionnaire itself) and the theoretical background behind the study. Moreover, we characterized the group in more detail and supported the results and discussion with more current literature (the references list almost doubled). We hope that now it is much more improved.

 In the Conclusion, the authors repeat the results from the Results and Discussion. For impact articles, the rule of thumb is - don't repeat anything! The last paragraph of the Conclusion is useful: students require help and support from the university, society and families and further: it would be useful to elaborate this message, not only in general terms, but really intertwined with the results.

Thank you for your valuable comment, we have corrected the conclusions section accordingly and we hope that now it was much more improved:

“Our results highlight that the mental health of university students in Poland is significantly affected in the face of a public health crisis. Students require help and support from the university, society and families. The positive outcome of the study was, that the physical activity may reduce loneliness, increase mental well-being and reduce the smoke tendencies. This study may be a suggestion for the government and universities to improve their collaboration with students in order to address mental health issues and related stress, loneliness, decreased motivation to study, and provide students with better quality and availability of psychological services, educational platforms, increase motivation for physical activity, and improve medical knowledge of students majoring in subjects not related to public health. Our findings help to specify the areas which disrupt the whole part of students’ lives and therefore may be used in the social campaigns fighting mental, social and financial side effects of COVID-19.”

With all these very useful comments we have heavily revised the manuscript and we wish that now it may be considered for publication in IJERPH. We are very grateful for all these valuable comments and help.

Reviewer 3 Report

The aim of the strudy is important and actual. Some remarks:

1./The number of students who answer the questionners should be higher, the sample size is low.

2./ The gender distribution of  participants is not recorded: how many female and male students participated?

3./ It is not mentioned, how many students of the lower grades and how many strudents of the advanced semesters were enrolled in the sudy ?

4./ A precise table is missing about  the number of students at the different university faculties (medicine, pharmacy, technical, agricultural, literature, etc. ?). The text on page 7, line 286 and page 8, line 287 is the following: "... a majority of our participants are from medical and health - related faculties". Absolute numbers and percentages are needed.

5./ The percentage of COVID vaccinated students is 39,3%. It is not mentioned  how many of them had the first, second or third vaccination?

6./ In many countries students of medical, dental and pharmacy faculties  had to be vaccinated almost in 100%, as well as the members of teaching staffs. The vaccination strategies could be different in different faculties of the Wroclav University - these data have to be presented.

7./ Not only sex is a "recreational activity" among university students, as it is written in the manuscript. There is no question about physical activities and sport - this group of questions is missing! The sentence on page 8, lines 314 - 315 is the following: " A positive aspect of the study was physical activity." No numerical data are proving this statement!

8./ International comparison of similar psychological study results gained during the pandemy is missing in the discussion part of the manuscript. The authors mention only one study from Jordan (page 7, line 243), no other ones.

9./ It is not mentioned, how many students were treated because of anxiety or depression either by psychotherapy or antidepressants.

10./ During the pandemy alcohol consumption increased and probably drug usage has been increased as well - even by students. Statistical data of this topic are missing in the study.

11./ "We are facing the upcoming 4th wave of COVID 19" (page 2, line 51) - is not a valid sentence any more , it has to be corrected. 

Basic completions of data and text are necessary. In this form the manuscript is not recommended for publication.

Author Response

Thank You for Your interesting observations. Here the original questions and answers to Your suggestions.

1./The number of students who answer the questionners should be higher, the sample size is low.

ANSWER 1 - Thank you very much for your observations, we discussed this aspect in the limitations of the study:

“The present study has a limitation - sample size for our survey (630) was not greater than the ones used in some other studies. However, smaller sample studies concerning similar problems, support our results, therefore we assume, our analyzed group may represent the general population relatively accurate. There have already been published studies on a similar size of population with reliable results [66–68]. First, our result of the decreased well-being of students is supported by Elmer’s study which revealed higher levels of stress, anxiety, loneliness, and depressive symptoms during the COVID-19 pandemic [66]. Study performed on about 250 Swiss individuals supported our result, claiming that the stress, anxiety, loneliness, and depressive symptoms get worsened during the COVID-19 pandemics [69]. Also, Alradhawi et al. in their review focus on students as the potentially susceptible group to disrupt their well-being by the disruption of daily routines [70]. The second factor, which worsened during the COVID-19 pandemics was the sexuality of the students. Study performed on 212 USA students revealed that the sexual activity of American students decreased dramatically for over 57% of reports [71]. Interestingly, study by Holland et al. warns that the level of sexual harassment in higher education has to be monitored even during the pandemic’s times [72]. Also, the narrative review by Eleuteri et al. focuses on the reduced sexual activity as the trigger to pornography consumption among young people [73]. Third problem of students are the concerns about financial status, which according to our studies increased among polish students. Similarly, study performed on New York city students (10000 sample size) obtained the same result as our study, showing, that the students experienced the anx-iety/depression and the financial instability due to the pandemics [74]. The last factor that was analyzed in this study was the trust in medical authorities. Here the highest dif-ferences between different countries-based studies are expected. Interestingly, the tendencies were similar too. For instance, most of the 1581 responders from the Romanian students survey had a positive attitude towards vaccination and similar tendency was observed among polish students [75]. In Saudi Arabia, most of 1232 participants complied to the precautionary practices to COVID-19 pandemics, which was generally observed among polish students as well [76].”

We hope that we proved that there are many studies that concern similar problems and obtain the reliable results. Moreover we discussed the similarities with the other studies, which proves that the conclusions of our study are universal.

2./ The gender distribution of  participants is not recorded: how many female and male students participated?

3./ It is not mentioned, how many students of the lower grades and how many strudents of the advanced semesters were enrolled in the sudy ?

4./ A precise table is missing about  the number of students at the different university faculties (medicine, pharmacy, technical, agricultural, literature, etc. ?). The text on page 7, line 286 and page 8, line 287 is the following: "... a majority of our participants are from medical and health - related faculties". Absolute numbers and percentages are needed.

ANSWERS 2,3,4 - As you suggested, we have included this information in a new table “Characteristics of students group” which is a table 2 now. We hope that we included all the required information.

5./ The percentage of COVID vaccinated students is 39,3%. It is not mentioned  how many of them had the first, second or third vaccination?

6./ In many countries students of medical, dental and pharmacy faculties  had to be vaccinated almost in 100%, as well as the members of teaching staffs. The vaccination strategies could be different in different faculties of the Wroclav University - these data have to be presented.

ANSWERS 5,6 – Thank you for the comment, we also think it is a very interesting question to include that in our study. However, we are currently working on a new project that aims to correlate the vaccination level of the students with the loss of COVID-19 related fear and we don’t want to overlap the results from both these studies. However, our point in this study was to assess the general attitude of the Polish students towards COVID-19 vaccination.

7./ Not only sex is a "recreational activity" among university students, as it is written in the manuscript. There is no question about physical activities and sport - this group of questions is missing! The sentence on page 8, lines 314 - 315 is the following: " A positive aspect of the study was physical activity." No numerical data are proving this statement!

ANSWER 7 – We also think that this is a very important aspect. Unfortunately, accidently we did not highlight these results in the first version of the manuscript. Now we have added the results and described them in the discussion. Our results show that physically active students were less likely to smoke and less likely to feel lonely. Data linked to physical activity have been included in the results and we have now broadened this aspect:

“Students who participated in the study were also asked about the ways of spending free time during the pandemic. A total of 34.1% of them spend it on physical activity and 45.9% spend it on hobbies. Respondents who declared to engage in a hobby (46.0%) seem to be also physically active (p=0.0300) and less interested in coming back to non-remote studies (p=0.0275). The presented study indicates that physical activity seems to have a positive impact on the mental health of students. Physically active students tended to smoke less (p=0.0497), and they felt less lonely (p=0.0083).”

And in the discussion:

“Similar tendencies have been noted in the British population, confirming that various interventions, such as learning a new hobby, can provide a distraction and help combat loneliness [41]. Several studies shows that during the constraints of the COVID 19 closure, physical activity was positively associated with psychological well-being [42–45]. These data are consistent with our findings, physically active students were less likely to smoke cigarettes and feel less lonely. During the pandemic, majority of university students from different countries had reduced levels of physical activity: walking, moderate, vigorous and total [46,47]. A study performed on Ukrainian students revealed that the physically inactive group had higher anxiety and depression levels than the physically active group [44]”

We hope that now the aspect of physical activity is more widely presented.

8./ International comparison of similar psychological study results gained during the pandemy is missing in the discussion part of the manuscript. The authors mention only one study from Jordan (page 7, line 243), no other ones.

ANSWER 8 - Thank you very much for this comment, certainly now that we have added a comparison of studies conducted in other countries, we hope that now our study will be more universal and professional:

“Similar tendencies have been noted in the British population, confirming that various interventions, such as learning a new hobby, can provide a distraction and help combat loneliness [41].”

And:

“During the pandemic, majority of university students from different countries had reduced levels of physical activity: walking, moderate, vigorous and total [46,47]. A study performed on Ukrainian students revealed that the physically inactive group had higher anxiety and depression levels than the physically active group [44].”

And further in the discussion:

“The present study has a limitation - sample size for our survey (630) was not greater than the ones used in some other studies. However, smaller sample studies concerning similar problems, support our results, therefore we assume, our analyzed group may represent the general population relatively accurate. There have already been published studies on a similar size of population with reliable results [66–68]. First, our result of the decreased well-being of students is supported by Elmer’s study which revealed higher levels of stress, anxiety, loneliness, and depressive symptoms during the COVID-19 pandemic [66]. Study performed on about 250 Swiss individuals supported our result, claiming that the stress, anxiety, loneliness, and depressive symptoms get worsened during the COVID-19 pandemics [69]. Also, Alradhawi et al. in their review focus on students as the potentially susceptible group to disrupt their well-being by the disruption of daily routines [70]. The second factor, which worsened during the COVID-19 pandemics was the sexuality of the students. Study performed on 212 USA students revealed that the sexual activity of American students decreased dramatically for over 57% of reports [71].Interestingly, study by Holland et al. warns that the level of sexual harassment in higher education has to be monitored even during the pandemic’s times [72]. Also, the narrative review by Eleuteri et al. focuses on the reduced sexual activity as the trigger to pornography consumption among young people [73]. Third problem of students are the concerns about financial status, which according to our studies increased among polish students. Similarly, study performed on New York city students (10000 sample size) obtained the same result as our study, showing, that the students experienced the anxiety/depression and the financial instability due to the pandemics [74]. The last factor that was analyzed in this study was the trust in medical authorities. Here the highest differences between different countries-based studies are expected. Interestingly, the tendencies were similar too. For instance, most of the 1581 responders from the Romanian students survey had a positive attitude towards vaccination and similar tendency was observed among polish students [75]. In Saudi Arabia, most of 1232 participants complied to the precautionary practices to COVID-19 pandemics, which was generally observed among polish students as well [76].”

We hope that now our results are well discussed and compared with the studies conducted in the other countries.

9./ It is not mentioned, how many students were treated because of anxiety or depression either by psychotherapy or antidepressants.

ANSWER 9 – Thank you for the comment, in our study we focused on the general population of the students and not the iatrogenic intervention to their health. However, we think the idea is very interesting and we will think about including that in our future research.

10./ During the pandemy alcohol consumption increased and probably drug usage has been increased as well - even by students. Statistical data of this topic are missing in the study.

ANSWER 10 – Thank you for the comment, we accidently missed the results in the first version of the manuscript. Now we have added the results in the 3.3 section:

“The meta-analysis of the data proves that apprehension about personal financial status increases the number of tobacco products smoked (p=0.0214) and the amount of alcohol consumed (p=0.0500). However, the increase of alcohol consumption in the general population was not statistically significant (p>0.05) after the pandemics started.”

 However, the use of drugs in Poland is not as prominent among the students (refer to the Polish government data https://www.cinn.gov.pl/portal?id=105923), that is why we did not include that in our questionnaire.  

11./ "We are facing the upcoming 4th wave of COVID 19" (page 2, line 51) - is not a valid sentence any more , it has to be corrected. 

Thank you for the comment we have updated the data up to the April 2022 stats.

With all these very useful comments we have heavily revised the manuscript and we wish that now it may be considered for publication in IJERPH. We are very grateful for all these valuable comments and help.

Round 2

Reviewer 2 Report

I congratulate the authors on how well they have been able to rework
and complete this article - all my comments have been accepted.
I recommend for adoption and publication.

Kateřina Ivanová

Reviewer 3 Report

The corrections and additional data are well incorporated, all these are accepted. The manuscript is recommended for publication.